# Input-Output Equivalence of Unitary and Contractive RNNs

**Melikasadat Emami**
Dept. ECE
UCLA
emami@ucla.edu

**Mojtaba Sahraee-Ardakan**
Dept. ECE
UCLA
msahraee@ucla.edu

**Sundeep Rangan**
Dept. ECE
NYU
srangan@nyu.edu

**Alyson K. Fletcher**
Dept. Statistics
UCLA
akfletcher@ucla.edu

## Abstract

Unitary recurrent neural networks (URNNs) have been proposed as a method to overcome the vanishing and exploding gradient problem in modeling data with long-term dependencies. A basic question is how restrictive is the unitary constraint on the possible input-output mappings of such a network? This work shows that for any contractive RNN with ReLU activations, there is a URNN with at most twice the number of hidden states and the identical input-output mapping. Hence, with ReLU activations, URNNs are as expressive as general RNNs. In contrast, for certain smooth activations, it is shown that the input-output mapping of an RNN cannot be matched with a URNN, even with an arbitrary number of states. The theoretical results are supported by experiments on modeling of slowly-varying dynamical systems.

## 1   Introduction

Recurrent neural networks (RNNs) – originally proposed in the late 1980s [20, 6] – refer to a widely-used and powerful class of models for time series and sequential data. In recent years, RNNs have become particularly important in speech recognition [9, 10] and natural language processing [5, 2, 24] tasks.

A well-known challenge in training recurrent neural networks is the *vanishing* and *exploding* gradient problem [3, 18]. RNNs have a transition matrix that maps the hidden state at one time to the next time. When the transition matrix has an induced norm greater than one, the RNN may become unstable. In this case, small perturbations of the input at some time can result in a change in the output that grows exponentially over the subsequent time. This instability leads to a so-called exploding gradient. Conversely, when the norm is less than one, perturbations can decay exponentially so inputs at one time have negligible effect in the distant future. As a result, the loss surface associated with RNNs can have steep walls that may be difficult to minimize. Such problems are particularly acute in systems with long-term dependencies, where the output sequence can depend strongly on the input sequence many time steps in the past.

Unitary RNNs (URNNs) [1] is a simple and commonly-used approach to mitigate the vanishing and exploding gradient problem. The basic idea is to restrict the transition matrix to be unitary (an orthogonal matrix for the real-valued case). The unitary transitional matrix is then combined with a non-expansive activation such as a ReLU or sigmoid. As a result, the overall transition mapping cannot amplify the hidden states, thereby eliminating the exploding gradient problem. In addition,

since all the singular values of a unitary matrix equal 1, the transition matrix does not attenuate the hidden state, potentially mitigating the vanishing gradient problem as well. (Due to activation, the hidden state may still be attenuated). Some early work in URNNs suggested that they could be more effective than other methods, such as long short-term memory (LSTM) architectures and standard RNNs, for certain learning tasks involving long-term dependencies [13, 1] – see a short summary below.

Although URNNs may improve the stability of the network for the purpose of optimization, a basic issue with URNNs is that the unitary contraint may potentially reduce the set of input-output mappings that the network can model. This paper seeks to rigorously characterize *how restrictive the unitary constraint is on an RNN*. We evaluate this restriction by comparing the set of input-output mappings achievable with URNNs with the set of mappings from all RNNs. As described below, we restrict our attention to RNNs that are contractive in order to avoid unstable systems.

We show three key results:

1. Given any contractive RNN with $n$ hidden states and ReLU activations, there exists a URNN with at most $2n$ hidden states and the identical input-ouput mapping.

2. This result is tight in the sense that, given any $n > 0$, there exists at least one contractive RNN such that any URNN with the same input-output mapping must have at least $2n$ states.

3. The equivalence of URNNs and RNNs depends on the activation. For example, we show that there exists a contractive RNN with *sigmoid* activations such that there is no URNN with any finite number of states that exactly matches the input-output mapping.

The implication of this result is that, for RNNs with ReLU activations, there is no loss in the *expressiveness* of model when imposing the unitary constraint. As we discuss below, the penalty is a two-fold increase in the number of parameters.

Of course, the expressiveness of a class of models is only one factor in their real performance. Based on these results alone, one cannot determine if URNNs will outperform RNNs in any particular task. Earlier works have found examples where URNNs offer some benefits over LSTMs and RNNs [1, 28]. But in the simulations below concerning modeling slowly-varying nonlinear dynamical systems, we see that URNNs with $2n$ states perform approximately equally to RNNs with $n$ states.

Theoretical results on generalization error are an active subject area in deep neural networks. Some measures of model complexity such as [17] are related to the spectral norm of the transition matrices. For RNNs with non-contractive matrices, these complexity bounds will grow exponentially with the number of time steps. In contrast, since unitary matrices can bound the generalization error, this work can also relate to generalizability.

**Prior work**

The vanishing and exploding gradient problem in RNNs has been known almost as early as RNNs themselves [3, 18]. It is part of a larger problem of training models that can capture long-term dependencies, and several proposed methods address this issue. Most approaches use some form of gate vectors to control the information flow inside the hidden states, the most widely-used being LSTM networks [11]. Other gated models include Highway networks [21] and gated recurrent units (GRUs) [4]. L1/L2 penalization on gradient norms and gradient clipping were proposed to solve the exploding gradient problem in [18]. With L1/L2 penalization, capturing long-term dependencies is still challenging since the regularization term quickly kills the information in the model. A more recent work [19] has successfully trained very deep networks by carefully adjusting the initial conditions to impose an approximate unitary structure of many layers.

Unitary evolution RNNs (URNNs) are a more recent approach first proposed in [1]. Orthogonal constraints were also considered in the context of associative memories [27]. One of the technical difficulties is to efficiently parametrize the set of unitary matrices. The numerical simulations in this work focus on relatively small networks, where the parameterization is not a significant computational issue. Nevertheless, for larger numbers of hidden states, several approaches have been proposed. The model in [1] parametrizes the transition matrix as a product of reflection, diagonal, permutation, and Fourier transform matrices. This model spans a subspace of the whole unitary space, thereby limiting the expressive power of RNNs. The work [28] overcomes this issue by optimizing over

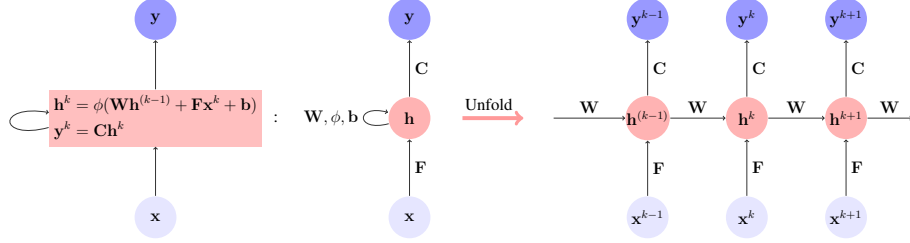

Figure 1: Recurrent Neural Network (RNN) model.

full-capacity unitary matrices. A key limitation in this work, however, is that the projection of weights on to the unitary space is not computationally efficient. A tunable, efficient parametrization of unitary matrices is proposed in [13]. This model provides the computational complexity of $O(1)$ per parameter. The unitary matrix is represented as a product of rotation matrices and a diagonal matrix. By grouping specific rotation matrices, the model provides tunability of the span of the unitary space and enables using different capacities for different tasks. Combining the parametrization in [13] for unitary matrices and the "forget" ability of the GRU structure, [4, 12] presented an architecture that outperforms conventional models in several long-term dependency tasks. Other methods such as orthogonal RNNs proposed by [16] showed that the unitary constraint is a special case of the orthogonal constraint. By representing an orthogonal matrix as a product of Householder reflectors, we are able span the entire space of orthogonal matrices. Imposing hard orthogonality constraints on the transition matrix limits the expressiveness of the model and speed of convergence and performance may degrade [26].

## 2   RNNs and Input-Output Equivalence

**RNNs.**   We consider recurrent neural networks (RNNs) representing sequence-to-sequence mappings of the form

$$\mathbf{h}^{(k)} = \phi(\mathbf{W}\mathbf{h}^{(k-1)} + \mathbf{F}\mathbf{x}^{(k)} + \mathbf{b}), \quad \mathbf{h}^{(-1)} = \mathbf{h}_{-1}, \tag{1a}$$

$$\mathbf{y}^{(k)} = \mathbf{C}\mathbf{h}^{(k)}, \tag{1b}$$

parameterized by $\boldsymbol{\Theta} = (\mathbf{W}, \mathbf{F}, \mathbf{b}, \mathbf{C}, \mathbf{h}_{-1})$. The system is shown in Fig. 1. The system maps a sequence of inputs $\mathbf{x}^{(k)} \in \mathbb{R}^m$, $k = 0, 1, \ldots, T-1$ to a sequence of outputs $\mathbf{y}^{(k)} \in \mathbb{R}^p$. In equation (1), $\phi$ is the activation function (e.g. sigmoid or ReLU); $\mathbf{h}^{(k)} \in \mathbb{R}^n$ is an internal or hidden state; $\mathbf{W} \in \mathbb{R}^{n \times n}, \mathbf{F} \in \mathbb{R}^{n \times m}$, and $\mathbf{C} \in \mathbb{R}^{p \times n}$ are the hidden-to-hidden, input-to-hidden, and hidden-to-output weight matrices respectively; and $\mathbf{b}$ is the bias vector. We have considered the initial condition, $\mathbf{h}_{-1}$, as part of the parameters, although we will often take $\mathbf{h}_{-1} = 0$. Given a set of parameters $\boldsymbol{\Theta}$, we will let

$$\mathbf{y} = G(\mathbf{x}, \boldsymbol{\Theta}) \tag{2}$$

denote the resulting sequence-to-sequence mapping. Note that the number of time samples, $T$, is fixed throughout our discussion.

Recall [23] that a matrix $\mathbf{W}$ is *unitary* if $\mathbf{W}^{\mathrm{H}}\mathbf{W} = \mathbf{W}\mathbf{W}^{\mathrm{H}} = \mathbf{I}$. When a unitary matrix is real-valued, it is also called *orthogonal*. In this work, we will restrict our attention to real-valued matrices, but still use the term unitary for consistency with the URNN literature. A *Unitary RNN* or URNN is simply an RNN (1) with a unitary state-to-state transition matrix $\mathbf{W}$. A key property of unitary matrices is that they are *norm-preserving*, meaning that $\|\mathbf{W}\mathbf{h}^{(k)}\|_2 = \|\mathbf{h}^{(k)}\|_2$. In the context of (1a), the unitary constraint implies that the transition matrix does not amplify the state.

**Equivalence of RNNs.**   Our goal is to understand the extent to which the unitary constraint in a URNN restricts the set of input-output mappings. To this end, we say that the RNNs for two parameters $\boldsymbol{\Theta}_1$ and $\boldsymbol{\Theta}_2$ are *input-output equivalent* if the sequence-to-sequence mappings are identical,

$$G(\mathbf{x}, \boldsymbol{\Theta}_1) = G(\mathbf{x}, \boldsymbol{\Theta}_2) \text{ for all } \mathbf{x} = (\mathbf{x}^{(0)}, \ldots, \mathbf{x}^{(T-1)}). \tag{3}$$

That is, for all input sequences $\mathbf{x}$, the two systems have the same output sequence. Note that the hidden internal states $\mathbf{h}^{(k)}$ in the two systems may be different. We will also say that two RNNs are *equivalent on a set of* $\mathcal{X}$ of inputs if (3) holds for all $\mathbf{x} \in \mathcal{X}$.

It is important to recognize that input-output equivalence does *not* imply that the parameters $\mathbf{\Theta}_1$ and $\mathbf{\Theta}_2$ are identical. For example, consider the case of linear RNNs where the activation in (1) is the identity, $\phi(\mathbf{z}) = \mathbf{z}$. Then, for any invertible $\mathbf{T}$, the transformation

$$\mathbf{W} \rightarrow \mathbf{T}\mathbf{W}\mathbf{T}^{-1}, \quad \mathbf{C} \rightarrow \mathbf{C}\mathbf{T}^{-1}, \quad \mathbf{F} \rightarrow \mathbf{T}\mathbf{F}, \quad \mathbf{h}_{-1} \rightarrow \mathbf{T}\mathbf{h}_{-1}, \tag{4}$$

results in the same input-output mapping. However, the internal states $\mathbf{h}^{(k)}$ will be mapped to $\mathbf{T}\mathbf{h}^{(k)}$. The fact that many parameters can lead to identical input-output mappings will be key to finding equivalent RNNs and URNNs.

**Contractive RNNs.** The *spectral norm* [23] of a matrix $\mathbf{W}$ is the maximum gain of the matrix $\|\mathbf{W}\| := \max_{\mathbf{h} \neq 0} \frac{\|\mathbf{W}\mathbf{h}\|_2}{\|\mathbf{h}\|_2}$. In an RNN (1), the spectral norm $\|\mathbf{W}\|$ measures how much the transition matrix can amplify the hidden state. For URNNs, $\|\mathbf{W}\| = 1$. We will say an RNN is *contractive* if $\|\mathbf{W}\| < 1$, *expansive* if $\|\mathbf{W}\| > 1$, and *non-expansive* if $\|\mathbf{W}\| \leq 1$. In the sequel, we will restrict our attention to contractive and non-expansive RNNs. In general, given an expansive RNN, we cannot expect to find an equivalent URNN. For example, suppose $\mathbf{h}^{(k)} = h^{(k)}$ is scalar. Then, the transition matrix $\mathbf{W}$ is also scalar $\mathbf{W} = w$ and $w$ is expansive if and only if $|w| > 1$. Now suppose the activation is a ReLU $\phi(h) = \max\{0, h\}$. Then, it is possible that a constant input $x^{(k)} = x_0$ can result in an output that grows exponentially with time: $y^{(k)} = \text{const} \times w^k$. Such an exponential increase is not possible with a URNN. We consider only non-expansive RNNs in the remainder of the paper. Some of our results will also need the assumption that the activation function $\phi(\cdot)$ in (1) is non-expansive:

$$\|\phi(\mathbf{x}) - \phi(\mathbf{y})\|_2 \leq \|\mathbf{x} - \mathbf{y}\|_2, \qquad \text{for all } \mathbf{x} \text{ and } \mathbf{y}.$$

This property is satisfied by the two most common activations, sigmoids and ReLUs.

**Equivalence of Linear RNNs.** To get an intuition of equivalence, it is useful to briefly review the concept in the case of linear systems [14]. Linear systems are RNNs (1) in the special case where the activation function is identity, $\phi(\mathbf{z}) = \mathbf{z}$; the initial condition is zero, $\mathbf{h}_{-1} = 0$; and the bias is zero, $\mathbf{b} = 0$. In this case, it is well-known that two systems are input-output equivalent if and only if they have the same *transfer function*,

$$H(s) := \mathbf{C}(s\mathbf{I} - \mathbf{W})^{-1}\mathbf{F}. \tag{5}$$

In the case of scalar inputs and outputs, $H(s)$ is a rational function of the complex variable $s$ with numerator and denominator degree of at most $n$, the dimension of the hidden state $\mathbf{h}^{(k)}$. Any state-space system (1) that achieves a particular transfer function is called a *realization* of the transfer function. Hence two linear systems are equivalent if and only if they are the realizations of the same transfer function.

A realization is called *minimal* if it is not equivalent some linear system with fewer hidden states. A basic property of realizations of linear systems is that they are minimal if and only if they are *controllable* and *observable*. The formal definition is in any linear systems text, e.g. [14]. Loosely, controllable implies that all internal states can be reached with an appropriate input and observable implies that all hidden states can be observed from the ouput. In absence of controllability and observability, some hidden states can be removed while maintaining input-output equivalence.

## 3 Equivalence Results for RNNs with ReLU Activations

Our first results consider contractive RNNs with ReLU activations. For the remainder of the section, we will restrict our attention to the case of zero initial conditions, $\mathbf{h}^{(-1)} = 0$ in (1).

**Theorem 3.1** *Let* $\mathbf{y} = G(\mathbf{x}, \mathbf{\Theta}_c)$ *be a contractive RNN with ReLU activation and states of dimension* $n$. *Fix* $M > 0$ *and let* $\mathcal{X}$ *be the set of all sequences such that* $\|\mathbf{x}^{(k)}\|_2 \leq M < \infty$ *for all* $k$. *Then there exists a URNN with state dimension* $2n$ *and parameters* $\mathbf{\Theta}_u = (\mathbf{W}_u, \mathbf{F}_u, \mathbf{b}_u, \mathbf{C}_u)$ *such that for all* $\mathbf{x} \in \mathcal{X}$, $G(\mathbf{x}, \mathbf{\Theta}_c) = G(\mathbf{x}, \mathbf{\Theta}_u)$. *Hence the input-output mapping is matched for bounded inputs.*

**Proof** See Appendix A.

Theorem 3.1 shows that for any contractive RNN with ReLU activations, there exists a URNN with at most twice the number of hidden states and the identical input-output mapping. Thus, there is no loss in the set of input-output mappings with URNNs relative to general contractive RNNs on bounded inputs.

The penalty for using RNNs is the two-fold increase in state dimension, which in turn increases the number of parameters to be learned. We can estimate this increase in parameters as follows: The raw number of parameters for an RNN (1) with $n$ hidden states, $p$ outputs and $m$ inputs is $n^2 + (p+m+1)n$. However, for ReLU activations, the RNNs are equivalent under the transformations (4) using diagonal positive $\mathbf{T}$. Hence, the number of degrees of freedom of a general RNN is at most $d_{\mathrm{rnn}} = n^2 + (p+m)n$. We can compare this value to a URNN with $2n$ hidden states. The set of $2n \times 2n$ unitary $\mathbf{W}$ has $2n(2n-1)/2$ degrees of freedom [22]. Hence, the total degrees of freedom in a URNN with $2n$ states is at most $d_{\mathrm{urnn}} = n(2n-1) + 2n(p+m)$. We conclude that a URNN with $2n$ hidden states has slightly fewer than twice the number of parameters as an RNN with $n$ hidden states.

We note that there are cases that the contractivity assumption is limiting, however, the limitations may not always be prohibitive. We will see in our experiments that imposing the contractivity constraint can improve learning for RNNs when models have sufficiently large numbers of time steps. Some related results where bounding the singular values help with the performance can be found in [26].

We next show a converse result.

**Theorem 3.2** *For every positive n, there exists a contractive RNN with ReLU nonlinearity and state dimension n such that every equivalent URNN has at least $2n$ states.*

**Proof** See Appendix B.1 in the Supplementary Material.

The result shows that the $2n$ achievability bound in Theorem 3.1 is tight, at least in the worst case. In addition, the RNN constructed in the proof of Theorem 3.2 is not particularly pathological. We will show in our simulations in Section 5 that URNNs typically need twice the number of hidden states to achieve comparable modeling error as an RNN.

## 4 Equivalence Results for RNNs with Sigmoid Activations

Equivalence between RNNs and URNNs depends on the particular activation. Our next result shows that with sigmoid activations, URNNs are, in general, never exactly equivalent to RNNs, even with an arbitrary number of states.

We need the following technical definition: Consider an RNN (1) with a standard sigmoid activation $\phi(z) = 1/(1 + e^{-z})$. If $\mathbf{W}$ is non-expansive, then a simple application of the contraction mapping principle shows that for any constant input $x^{(k)} = x^*$, there is a fixed point in the hidden state $\mathbf{h}^* = \phi(\mathbf{W}\mathbf{h}^* + \mathbf{F}\mathbf{x}^* + \mathbf{b})$. We will say that the RNN is controllable and observable at $\mathbf{x}^*$ if the linearization of the RNN around $(\mathbf{x}^*, \mathbf{h}^*)$ is controllable and observable.

**Theorem 4.1** *There exists a contractive RNN with sigmoid activation function $\phi$ with the following property: If a URNN is controllable and observable at any point $\mathbf{x}^*$, then the URNN cannot be equivalent to the RNN for inputs $\mathbf{x}$ in the neighborhood of $\mathbf{x}^*$.*

**Proof** See Appendix B.2 in the Supplementary Material.

The result provides a converse on equivalence: Contractive RNNs with sigmoid activations are not in general equivalent to URNNs, even if we allow the URNN to have an arbitrary number of hidden states. Of course, the approximation error between the URNN and RNN may go to zero as the URNN hidden dimension goes to infinity (e.g., similar to the approximation results in [8]). However, exact equivalence is not possible with sigmoid activations, unlike with ReLU activations. Thus, there is fundamental difference in equivalence for smooth and non-smooth activations.

We note that the fundamental distinction between Theorem 3.1 and the opposite result in Theorem 4.1 is that the activation is smooth with a positive slope. With such activations, you can linearize the

system, and the eigenvalues of the transition matrix become visible in the input-output mapping. In contrast, ReLUs can zero out states and suppress these eigenvalues. This is a key insight of the paper and a further contribution in understanding nonlinear systems.

## 5   Numerical Simulations

In this section, we numerically compare the modeling ability of RNNs and URNNs where the true system is a contractive RNN with long-term dependencies. Specifically, we generate data from multiple instances of a synthetic RNN where the parameters in (1) are randomly generated. For the true system, we use $m = 2$ input units, $p = 2$ output units, and $n = 4$ hidden units at each time step. The matrices $\mathbf{F}$, $\mathbf{C}$ and $\mathbf{b}$ are generated as i.i.d. Gaussians. We use a random transition matrix,

$$\mathbf{W} = \mathbf{I} - \epsilon \mathbf{A}^\mathsf{T} \mathbf{A} / \|\mathbf{A}\|^2, \tag{6}$$

where $\mathbf{A}$ is Gaussian i.i.d. matrix and $\epsilon$ is a small value, taken here to be $\epsilon = 0.01$. The matrix (6) will be contractive with singular values in $(1 - \epsilon, 1)$. By making $\epsilon$ small, the states of the system will vary slowly, hence creating long-term dependencies. In analogy with linear systems, the time constant will be approximately $1/\epsilon = 100$ time steps. We use ReLU activations. To avoid degenerate cases where the outputs are always zero, the biases $\mathbf{b}$ are adjusted to ensure that the each hidden state is on some target $60\%$ of the time using a similar procedure as in [7].

The trials have $T = 1000$ time steps, which corresponds to 10 times the time constant $1/\epsilon = 100$ of the system. We added noise to the output of this system such that the signal-to-noise ratio (SNR) is 15 dB or 20 dB. In each trial, we generate 700 training samples and 300 test sequences from this system.

Given the input and the output data of this contractive RNN, we attempt to learn the system with: (i) standard RNNs, (ii) URNNs, and (iii) LSTMs. The hidden states in the model are varied in the range $n = [2, 4, 6, 8, 10, 12, 14]$, which include values both above and below the true number of hidden states $n_{\text{true}} = 4$. We used mean-squared error as the loss function. Optimization is performed using Adam [15] optimization with a batch size = 10 and learning rate = 0.01. All models are implemented in the Keras package in Tensorflow. The experiments are done over 30 realizations of the original contractive system.

For the URNN learning, of all the proposed algorithms for enforcing the unitary constraints on transition matrices during training [13, 28, 1, 16], we chose to project the transition matrix on the full space of unitary matrices after each iteration using singular value decomposition (SVD). Although SVD requires $\mathcal{O}(n^3)$ computation for each projection, for our choices of hidden states it performed faster than the aforementioned methods.

Since we have training noise and since optimization algorithms can get stuck in local minima, we cannot expect "exact" equivalence between the learned model and true system as in the theorems. So, instead, we look at the test error as a measure of the closeness of the learned model to the true system. Figure 2 on the left shows the test $R^2$ for a Gaussian i.i.d. input and output with SNR = 20 dB for RNNs, URNNs, and LSTMs. The red dashed line corresponds to the optimal $R^2$ achievable at the given noise level.

Note that even though the true RNN has $n_{\text{true}} = 4$ hidden states, the RNN model does not obtain the optimal test $R^2$ at $n = 4$. This is not due to training noise, since the RNN is able to capture the full dynamics when we over-parametrize the system to $n \approx 8$ hidden states. The test error in the RNN at lower numbers of hidden states is likely due to the optimization being caught in a local minima.

What is important for this work though is to compare the URNN test error with that of the RNN. We observe that URNN requires approximately twice the number of hidden states to obtain the same test error as achieved by an RNN. To make this clear, the right plot shows the same performance data with number of states adjusted for URNN. Since our theory indicates that a URNN with $2n$ hidden states is as powerful as an RNN with $n$ hidden states, we compare a URNN with $2n$ hidden units directly with an RNN with $n$ hidden units. We call this the adjusted hidden units. We see that the URNN and RNN have similar test error when we appropriately scale the number of hidden units as predicted by the theory.

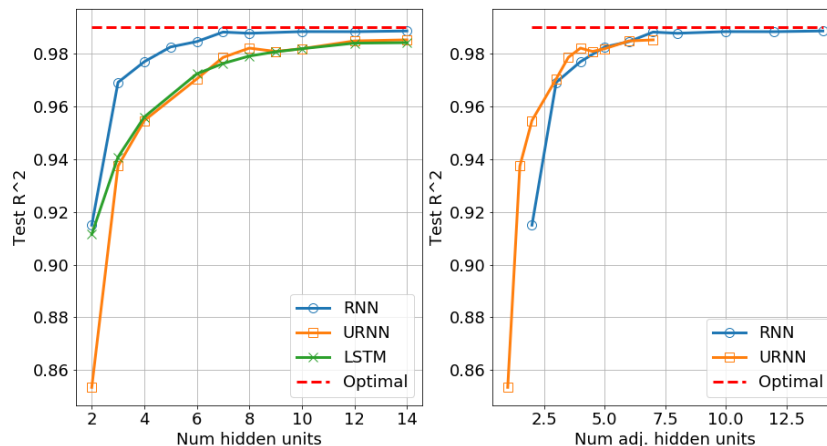

Figure 2: Test $R^2$ on synthetic data for a Gaussian i.i.d. input and output SNR=20 dB.

For completeness, the left plot in Figure 2 also shows the test error with an LSTM. It is important to note that the URNN has almost the same performance as an LSTM with considerably smaller number of parameters.

Figure 3 shows similar results for the same task with SNR = 15 dB. For this task, the input is *sparse* Gaussian i.i.d., i.e. Gaussian with some probability $p = 0.02$ and $0$ with probability $1 - p$. The left plot shows the $R^2$ vs. the number of hidden units for RNNs and URNNs and the right plot shows the same results once the number of hidden units for URNN is adjusted.

We also compared the modeling ability of URNNs and RNNs using the Pixel-Permuted MNIST task. Each MNIST image is a $28 \times 28$ grayscale image with a label between 0 and 9. A fixed random permutation is applied to the pixels and each pixel is fed to the network in each time step as the input and the output is the predicted label for each image [1, 13, 26].

We evaluated various models on the Pixel-Permuted MNIST task using validation based early stopping. Without imposing a contractivity constraint during learning, the RNN is either unstable or requires a slow learning rate. Imposing a contractivity constraint improves the performance. Incidentally, using a URNN improves the performance further. Thus, contractivity can improve learning for RNNs when models have sufficiently large numbers of time steps.

## 6   Conclusion

Several works empirically show that using unitary recurrent neural networks improves the stability and performance of the RNNs. In this work, we study how restrictive it is to use URNNs instead of RNNs. We show that URNNs are at least as powerful as contractive RNNs in modeling input-output mappings if enough hidden units are used. More specifically, for any contractive RNN we explicitly construct a URNN with twice the number of states of the RNN and identical input-output mapping. We also provide converse results for the number of state and the activation function needed for exact matching. We emphasize that although it has been shown that URNNs outperform standard RNNs and LSTM in many tasks that involve long-term dependencies, our main goal in this paper is to show that from an approximation viewpoint, URNNs are as expressive as general contractive RNNs. By a two-fold increase in the number of parameters, we can use the stability benefits they bring for optimization of neural networks.

## Acknowledgements

The work of M. Emami, M. Sahraee-Ardakan, A. K. Fletcher was supported in part by the National Science Foundation under Grants 1254204 and 1738286, and the Office of Naval Research under

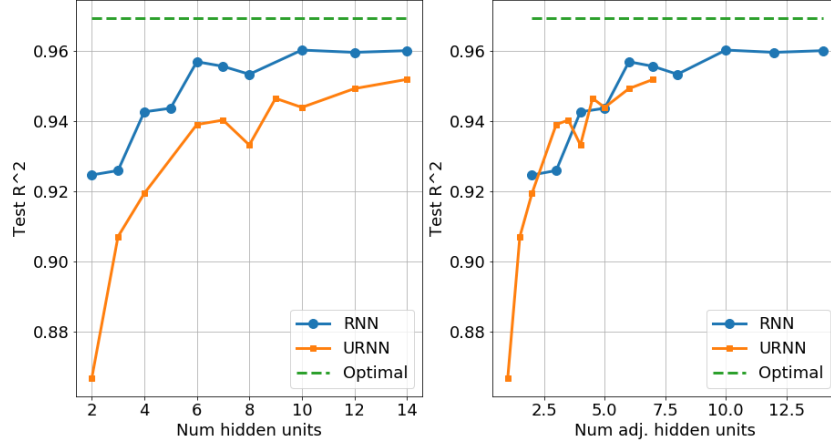

Figure 3: Test $R^2$ on synthetic data for a Gaussian i.i.d. input and output SNR=15 dB.

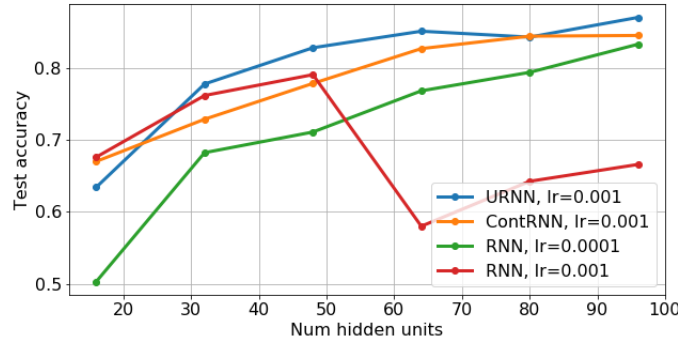

Figure 4: Accuracy on Permuted MNIST task for various models trained with RMSProp, validation-based early termination, and initial learning rate `lr`. (1) URNN model: RNN model with unitary constraint; (2) ContRNN: RNN with a contractivity constraint; (3 & 4) RNN model with no contractivity or unitary constraint (two learning rates). We see contractivity improves performance, and unitary constraints improve performance further.

Grant N00014-15-1-2677. S. Rangan was supported in part by the National Science Foundation under Grants 1116589, 1302336, and 1547332, NIST, the industrial affiliates of NYU WIRELESS, and the SRC.

## A Proof of Theorem 3.1

The basic idea is to construct a URNN with $2n$ states such that first $n$ states match the states of RNN and the last $n$ states are always zero. To this end, consider any contractive RNN,

$$\mathbf{h}_c^{(k)} = \phi(\mathbf{W}_c \mathbf{h}_c^{(k-1)} + \mathbf{F}_c \mathbf{x}^{(k)} + \mathbf{b}_c), \quad \mathbf{y}^{(k)} = \mathbf{C}_c \mathbf{h}_c^{(k)},$$

where $\mathbf{h}^{(k)} \in \mathbb{R}^n$. Since $\mathbf{W}$ is contractive, we have $\|\mathbf{W}\| \le \rho$ for some $\rho < 1$. Also, for a ReLU activation, $\|\phi(\mathbf{z})\| \le \|\mathbf{z}\|$ for all pre-activation inputs $\mathbf{z}$. Hence,

$$\|\mathbf{h}_c^{(k)}\|_2 = \|\phi(\mathbf{W}_c \mathbf{h}_c^{(k-1)} + \mathbf{F}_c \mathbf{x}^{(k)} + \mathbf{b}_c)\|_2 \le \|\mathbf{W}_c \mathbf{h}_c^{(k-1)} + \mathbf{F}_c \mathbf{x}^{(k)} + \mathbf{b}_c\|_2$$
$$\le \rho \|\mathbf{h}_c^{(k-1)}\|_2 + \|\mathbf{F}_c\| \|\mathbf{x}^{(k)}\|_2 + \|\mathbf{b}_c\|_2.$$

Therefore, with bounded inputs, $\|\mathbf{x}^{(k)}\| \leq M$, we have the state is bounded,

$$\|\mathbf{h}^{(k)}\|_2 \leq \frac{1}{1-\rho} \left[\|\mathbf{F}_c\|M + \|\mathbf{b}_c\|_2\right] =: M_h. \tag{7}$$

We construct a URNN as,

$$\mathbf{h}_u^{(k)} = \phi(\mathbf{W}_u \mathbf{h}_u^{(k-1)} + \mathbf{F}_u \mathbf{x}^{(k)} + \mathbf{b}_u), \quad \mathbf{y}^{(k)} = \mathbf{C}_u \mathbf{h}_u^{(k)}$$

where the parameters are of the form,

$$\mathbf{h}_u = \begin{bmatrix} \mathbf{h}_1 \\ \mathbf{h}_2 \end{bmatrix} \in \mathbb{R}^{2n}, \quad \mathbf{W}_u = \begin{bmatrix} \mathbf{W}_1, \mathbf{W}_2 \\ \mathbf{W}_3, \mathbf{W}_4 \end{bmatrix}, \quad \mathbf{F}_u = \begin{bmatrix} \mathbf{F}_c \\ \mathbf{0} \end{bmatrix}, \quad \mathbf{b}_u = \begin{bmatrix} \mathbf{b}_c \\ \mathbf{b}_2 \end{bmatrix}. \tag{8}$$

Let $\mathbf{W}_1 = \mathbf{W}_c$. Since $\|\mathbf{W}_c\| < 1$, we have $\mathbf{I} - \mathbf{W}_c^\mathsf{T}\mathbf{W}_c \succeq 0$. Therefore, there exists $\mathbf{W}_3$ such that $\mathbf{W}_3^\mathsf{T}\mathbf{W}_3 = \mathbf{I} - \mathbf{W}_c^\mathsf{T}\mathbf{W}_c$. With this choice of $\mathbf{W}_3$, the first $n$ columns of $\mathbf{W}_u$ are orthonormal. Let $\begin{bmatrix} \mathbf{W}_2 \\ \mathbf{W}_4 \end{bmatrix}$ extend these to an orthonormal basis for $\mathbb{R}^{2n}$. Then, the matrix $\mathbf{W}_u$ will be orthonormal.

Next, let $\mathbf{b}_2 = -M_h \mathbf{1}_{n \times 1}$, where $M_h$ is defined in (7). We show by induction that for all $k$,

$$\mathbf{h}_1^{(k)} = \mathbf{h}_c^{(k)}, \quad \mathbf{h}_2^{(k)} = \mathbf{0}. \tag{9}$$

If both systems are initialized at zero, (9) is satisfied at $k = -1$. Now, suppose this holds up to time $k - 1$. Then,

$$\begin{aligned} \mathbf{h}_1^{(k)} &= \phi(\mathbf{W}_1 \mathbf{h}_1^{(k-1)} + \mathbf{W}_2 \mathbf{h}_2^{(k-1)} + \mathbf{F}_c \mathbf{x}^{(k)} + \mathbf{b}_c) \\ &= \phi(\mathbf{W}_1 \mathbf{h}_1^{(k-1)} + \mathbf{F}_c \mathbf{x}^{(k)} + \mathbf{b}_c) = \mathbf{h}_c^{(k)}, \end{aligned}$$

where we have used the induction hypothesis that $\mathbf{h}_2^{(k-1)} = \mathbf{0}$. For $\mathbf{h}_2^{(k)}$, note that

$$\|\mathbf{W}_3 \mathbf{h}_1^{(k-1)}\|_\infty \leq \|\mathbf{W}_3 \mathbf{h}_1^{(k-1)}\|_2 \leq \|\mathbf{h}_1^{(k-1)}\| \leq M_h, \tag{10}$$

where the last step follows from (7). Therefore,

$$\mathbf{W}_3 \mathbf{h}_1^{(k-1)} + \mathbf{W}_4 \mathbf{h}_2^{(k-1)} + \mathbf{b}_2 = \mathbf{W}_3 \mathbf{h}_1^{(k-1)} - M \mathbf{1}_{n \times 1} \leq \mathbf{0}. \tag{11}$$

Hence with ReLU activation $\mathbf{h}_2^{(k)} = \phi(\mathbf{W}_3 \mathbf{h}_1^{(k-1)} + \mathbf{W}_4 \mathbf{h}_2^{(k-1)} + \mathbf{b}_2) = \mathbf{0}$. By induction, (9) holds for all $k$. Then, if we define $\mathbf{C}_u = [\mathbf{C}_c \mathbf{0}]$, we have the output of the URNN and RNN systems are identical

$$\mathbf{y}_u^{(k)} = \mathbf{C}_u \mathbf{h}_u^{(k)} = \mathbf{C}_c \mathbf{h}_1^{(k)} = \mathbf{y}_c^{(k)}.$$

This shows that the systems are equivalent.

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
