[Supplementary Material]

# Supplementary Material

 ## B    Converse Theorem Proofs

 ### B.1    Proof of Theorem 3.2

 First consider the case when $n = 1$ with scalar inputs and outputs. Let $\theta_c = (w_c, f_c, b_c, c_c)$ be the
 parameters of a contractive RNN with $f_c = c_c = 1$, $b_c = 0$ and $w_c \in (0, 1)$. Hence, the contractive
 RNN is given by

$$h_c^{(k)} = \phi(w_c h_c^{(k-1)} + x^{(k)}), \quad y^{(k)} = h_c^{(k)}, \tag{12}$$

 and $\phi(z) = \max\{0, z\}$ is the ReLU activation. Suppose $\boldsymbol{\Theta}_u$ are the parameters of an equivalent
 URNN. If $\boldsymbol{\Theta}$ has less than $2n = 2$ states, it must have $n = 1$ state. Let the equivalent URNN be

$$h_u^{(k)} = \phi(w_u h_u^{(k-1)} + f_u x^{(k)} + b_u), \quad y^{(k)} = c_u h_u^{(k)}, \tag{13}$$

 for some parameters $\boldsymbol{\Theta}_u = (w_u, f_u, b_u, c_u)$. Since $w_u$ is orthogonal, either $w_u = 1$ or $w_u = -1$.
 Also, either $f_u > 0$ or $f_u < 0$. First, consider the case when $w_u = 1$ and $f_u > 0$. Then, there exists
 a large enough input $x^{(k)}$ such that for all time steps $k$, both systems are operating in the active phase
 of ReLU. Therefore, we have two equivalent linear systems,

$$\text{contractive RNN:} \quad h_c^{(k)} = w_c h_c^{(k-1)} + x^{(k)}, \quad y^{(k)} = h_c^{(k)} \tag{14}$$

$$\text{URNN:} \quad h_u^{(k)} = h_u^{(k-1)} + f_u x^{(k)} + b_u, \quad y^{(k)} = c_u h_u^{(k)}. \tag{15}$$

 In order to have identical input-output mapping for these linear systems for all $x$, it is required that
 $w_c = 1$, which is a contradiction. The other cases $w_c = -1$ and $f_u < 0$ can be treated similarly.
 Therefore, at least $n = 2$ states are needed for the URNN to match the contractive RNN with $n = 1$
 state.

 For the case of general $n$, consider the contractive RNN,

$$\mathbf{h}^{(k)} = \phi(\mathbf{W}\mathbf{h}^{(k-1)} + \mathbf{F}\mathbf{x}^{(k)} + \mathbf{b}), \quad \mathbf{y}^{(k)} = \mathbf{C}\mathbf{h}^{(k)}, \tag{16}$$

 where $\mathbf{W} = \text{diag}(w_c, w_c, ..., w_c)$, $\mathbf{F} = \text{diag}(f_c, f_c, ..., f_c)$, $\mathbf{b} = b_c \mathbf{1}_{n \times 1}$, and $\mathbf{C} =$
 $\text{diag}(c_c, c_c, ..., c_c)$. This system is separable in that if $\mathbf{y} = G(\mathbf{x})$ then $y_i = G(x_i, \theta_c)$ for each
 input $i$. A URNN system will need 2 states for each scalar system requiring a total of $2n$ states.

 ### B.2    Proof of Theorem 4.1

 We use the same scalar contractive RNN (12), but with a sigmoid activation $\phi(z) = 1/(1 + e^{-z})$.
 Let $\boldsymbol{\Theta} = (\mathbf{W}_u, \mathbf{f}_u, \mathbf{c}_u, \mathbf{b}_u)$ be the parameters of any URNN with scalar input and outputs. Suppose
 the URNN is controllable and observable at an input value $x^*$. Let $h_c^*$ and $\mathbf{h}_u^*$ be, respectively, the
 fixed points of the hidden states for the contractive RNN and URNN:

$$\text{contractive RNN:} \quad h_c^* = \phi(w_c h_c + x^*), \tag{17}$$

$$\text{URNN:} \quad \mathbf{h}_u^* = \phi(\mathbf{W}_u \mathbf{h}_u + \mathbf{f}_u x^* + \mathbf{b}_u). \tag{18}$$

 We take the linearizations [24] of each system around its fixed point and apply a small perturbation
 $\Delta x$ around $x^*$. Therefore, we have two linear systems with identical input-output mapping given by,

$$\text{contractive RNN:} \quad \Delta h_c^{(k)} = d_c(w_c \Delta h^{(k-1)} + \Delta x^{(k)}), \quad y^{(k)} = \Delta h_c^{(k)} + h_c^*, \tag{19}$$

$$\text{URNN:} \quad \Delta \mathbf{h}_u^{(k)} = \mathbf{D}_u(\mathbf{W}_u \Delta \mathbf{h}_u^{(k-1)} + \mathbf{f}_u^\mathsf{T} \Delta x^{(k)}), \quad y^{(k)} = \mathbf{c}_u^\mathsf{T} \Delta \mathbf{h}_u + \mathbf{c}_u^\mathsf{T} \mathbf{h}_u^*, \tag{20}$$

 where

$$d_c = \phi'(z_c^* = w_c h_c^* + x^*), \quad \mathbf{D}_u = \phi'(\mathbf{W}_u \mathbf{h}_u^* + \mathbf{f}_u x^* + \mathbf{b}_u),$$

 are the derivatives of the activations at the fixed points. Since both systems are controllable and
 observable, their dimensions must be the same and the eigenvalues of the transition matrix must
 match. In particular, the URNN must be scalar, so $\mathbf{W}_u = w_u$ for some scalar $w_u$. For orthogonality,
 either $w_u = 1$ or $w_u = -1$. We look at the $w_u = 1$ case; the $w_u = -1$ case is similar. Since the
 eigenvalues of the transition matrix must match we have,

$$d_c w_c = d_u \Rightarrow \phi'(w_c h_c^* + x^*) w_c = \phi'(h_u^* + f_u x^* + b_u). \tag{21}$$

where $h_u^*$ and $h_c^*$ are the solutions to the fixed point equations:

$$h_c^* = \phi(w_c h_c + x^*), \quad h_u^* = \phi(h_u^* + f_b x^* + b_u). \tag{22}$$

Also, since two systems have the same output,

$$h_c^* = c_u h_u^*. \tag{23}$$

Now, (21) must hold at any input $x^*$ where the URNN is controllable and observable. If the URNN is controllable and observable at some $x^*$, it is is controllable and observable in a neighborhood of $x^*$. Hence, (21) and (23) holds in some neighborhood of $x^*$. To write this mathematically, define the functions,

$$g_c(x^*) := \begin{bmatrix} w_c \phi'(w_c h_c^* + x^*) \\ h_c^* \end{bmatrix}, \quad g_u(x^*) := \begin{bmatrix} \phi'(h_u^* + f_u x^* + b_u) \\ c_u h_u^* \end{bmatrix}, \tag{24}$$

where, for a given $x^*$, $h_u^*$ and $h_c^*$ are the solutions to the fixed point equations (22). We must have that $g_c(x^*) = g_u^*(x^*)$ for all $x^*$ in some neighborhood. Taking derivatives of (24) and using the fact that $\phi(z)$ being a sigmoid, one can show that this matching can only occur when,

$$w_c = 1, \quad b_u = 0, \quad c_u = 1.$$

This is a contradiction since we have assumed that the RNN system is contractive which requires $|w_c| = 1$.