[Reviews · NeurIPS 2019]

Reviewer 1



UPDATE: I’m largely happy with how the authors addressed my points. I still think that the requirement for RNN to be non-expansive is quite restrictive per se, but this work may still be a good starting point for further theoretical discussion of such issues. The authors provide a straightforward proof by construction that a URNN with two times the number of hidden states as the corresponding RNN is as expressive as the RNN, i.e. can be formulated such that it produces the same outputs for the same series of inputs. While this is true for RNN with ReLU activation, the authors further prove, by linearizing around fixed points, that this is generally not true for RNN/URNN with sigmoid activation. Strengths: - Given that URNN are an important technique for modeling long-term dependencies, while avoiding some of the complexities of LSTM/GRU, rigorous theoretical results on how restrictive the unitary constraint is are timely and important. As far as I’m aware, this is the first set of such results. Weaknesses: - The proof works only under the assumption that the corresponding RNN is contractive, i.e. has no diverging directions in its eigenspace. As the authors point out (line #127), for expansive RNN there will usually be no corresponding URNN. While this is true, I think it still imposes a strong limitation a priori on the classes of problems that could be computed by an URNN. For instance chaotic attractors with at least one diverging eigendirection are ruled out to begin with. I think this needs further discussion. For instance, could URNN/ contractive RNN still *efficiently* solve some of the classical long-term RNN benchmarks, like the multiplication problem? Minor stuff: - Statement on line 134: Only true for standard sigmoid [1+exp(-x)]^-1, depends on max. slope - Theorem 4.1: Would be useful to elaborate a bit more in the main text why this holds (intuitively, since the RNN unlike the URNN will converge to the nearest FP). - line 199: The difference is not fundamental but only for the specific class of smooth (sigmoid) and non-smooth (ReLU) activation functions considered I think? Moreover: Is smoothness the crucial difference at all, or rather the fact that sigmoid is truly contractive while ReLU is just non-expansive? - line 223-245: Are URNN at all practical given the costly requirement to enforce the unitary matrix after each iteration?

Reviewer 2



Overall the paper focuses on theoretic analysis of the expressive powers of RNN's in terms of generating a desired sequence, but does not provide any implementable strategies to improve over existing algorithms in terms of avoiding vanishing or explosive gradient. Another concern is that generating desired output sequence'' may not be directly related to the generalization capacity of RNN in terms of predicting future signals, and so it would be more desirable if the analysis can bridge this gap.

Reviewer 3



Originality: To my knowledge the results in this work are clearly new and interesting. They build on and advance existing works. Quality: The paper appears to be a complete, and self-contained work that backs up claims with proofs or results. The paper states both what can be achieved but also what cannot. Clarity: The paper is well written and organised. It introduces the problem very well, and all relevant terms are well introduced. The supplementary material contains some of the proofs for theorems in the main paper. Significance: I believe the results of this work are important for future research of RNN and their training methods. While earlier work already looked into orthogonal networks (mostly for memory capacity, eg White, O, Lee, D, and Sompolinky, H. Short-term memory in orthogonal neural networks; also others Mikael Henaff et al Recurrent Orthogonal Networks and Long-Memory Tasks), expressiveness of the approaches has not been compared in this form, at least to my knowledge.

[Author Response · NeurIPS 2019]

**Response to Reviewers.** We would like to thank the reviewers for their valuable feedback. All the reviewers recognized that the paper made novel theoretical contributions for an important class of models for which there have been few such results. We appreciate and address the reviewers' suggestions for improvement as follows.

**Reviewer 1**: We are glad the reviewer found the paper to be timely and important with rigorous and novel theoretical results on an important class of models for problems with long-term dependencies.

*Contractive assumption*: We agree that there are cases where this key assumption is limiting. Indeed, as stated in the paper, a URNN cannot in general be equivalent to an unstable system. However, the contractivity assumption is not always prohibitively limiting. For example, we are excited to address the reviewer's concern of comparison to a benchmark. In Fig. 1 below, we now evaluate various models on the standard permuted MNIST task (see [1, 13, 25] of the paper) using validation-based early stopping. Permuted MNIST is a more widely-used benchmark for this class of problems than the multiplication task. Without imposing a contractivity constraint during learning, the RNN is either unstable or requires a slow learning rate. Imposing a contractivity constraint improves the performance. Incidentally, using a URNN improves the performance further. Thus, contractivity can improve learning for RNNs when models have sufficiently large numbers of time steps. Related results, where bounding the singular values can help, are found in [25] of the paper. We will include these experiments and discussion in the final paper. Thank you for raising this issue.

Figure 1: Accuracy on Permuted MNIST task for various models trained with RMSProp, validation-based early termination, and initial learning rate `lr`. (1) URNN model: RNN model with unitary constraint; (2) ContRNN: RNN with a contractivity constraint; (3 & 4) RNN model with no contractivity or unitary constraint (two learning rates). We see contractivity improves performance, and unitary constraints improve performance further.

*Other concerns:* (1) The reviewer is correct that the result requires the standard sigmoid; we will state this. It can also be extended to other smooth activations with slope $< 1$. (2) The fixed points exist for the URNN since the activation slope is $< 1$. (3) The reviewer is correct that the fundamental distinction between Theorem 3.1 and the converse result 4.1 is that the activation is smooth with a positive slope. With such activations, you can linearize the system, and the eigenvalues of the transition matrix become visible in the input–output mapping. In contrast, ReLUs can zero out states and suppress these eigenvalues. This is a key insight of the paper and a further contribution in understanding nonlinear systems. (4) There are several algorithms [1, 13, 16, 25, 26] for efficiently implementing the unitary constraint.

**Reviewer 2**: *Connection to algorithms*: The reviewer is correct that the focus of the work was on theoretical properties of existing models and algorithms. Since there are already many works on efficient algorithms (see [1, 13, 16, 25, 26] of the paper) but few methods to analyze them, this direction would be more impactful. As Reviewer 3 noted, we believe that our theory can guide future algorithms. For example, much work (e.g., [13] and [17] in the paper) developed efficient parametrizations of the matrices, some covering only a subset of unitary space. The results in this paper may lead to better understanding and improved efficient representations. In particular, for representations, coverage of input–output relationships is more important than coverage of the space of transition matrices. Our results suggest that even more efficient representations are possible if we parametrize the set of input–output mappings.

*Generalization error*: Reviewer 2 is correct that expressivity is only one component of generalization error. Theoretical results on generalization error are a difficult and active subject area in deep neural networks. However, some measures of model complexity such as in [A] are related to the spectral norm of the transition matrices. For RNNs with non-contractive matrices, these complexity bounds will grow exponentially with the number of time steps. In contrast, since unitary matrices can bound the generalization error, our updated work can also relate to generalizability. Thank you for raising this important issue. We have already added this valuable and new result and discussion.

**Reviewer 3**: We are glad the reviewer found the work complete and self-contained, with backed-up claims and clear statements of what can and cannot be achieved. We agree the work goes beyond the memory capacity of orthogonal networks analysis by White, Lee, and Sompolinsky [B]. In particular, we develop a novel approach for formalizing and analyzing input–output expressiveness, which was not previously examined. Thank you for this reference; we will add this. We too hope that this result is important for future research of RNNs and developing training methods.

[A] Neyshabur, B., Bhojanapalli, S., McAllester, D. and Srebro, N., 2017. Exploring generalization in deep learning. In *Proc. NIPS*.

[B] White, O.L., Lee, D.D. and Sompolinsky, H., 2004. Short-term memory in orthogonal neural networks. *Physical Review Letters*, 92(14).


[Meta-Review · NeurIPS 2019]

Although the paper does not suggest any specific direction for designing better RNN architectures, it provides relevant and significant theoretical results on the expressivity of Unitary RNN. Thus it can be stated that the paper contributes to a better understanding of the capabilities of Unitary RNN and for this reason is worth to be accepted. The rebuttal contributed to clarify issues raised by reviewers.